# Mixing Genetically and Morphologically Distinct Populations in Translocations: Asymmetrical Introgression in A Newly Established Population of the Boodie (*Bettongia lesueur*)

**DOI:** 10.3390/genes10090729

**Published:** 2019-09-19

**Authors:** Rujiporn Thavornkanlapachai, Harriet R. Mills, Kym Ottewell, Judy Dunlop, Colleen Sims, Keith Morris, Felicity Donaldson, W. Jason Kennington

**Affiliations:** 1School of Biological Sciences, The University of Western Australia, Crawley, Western Australia 6009, Australia; jason.kennington@uwa.edu.au; 2Centre for Ecosystem Management, School of Science, Edith Cowan University, Joondalup, Western Australia 6027, Australia; harriet.mills@ecu.edu.au; 3Department of Biodiversity, Conservation and Attractions, Locked Bag 104, Bentley Delivery Centre, Western Australia 6152, Australia; kym.ottewell@dbca.wa.gov.au; 4School of Veterinary and Biomedical Sciences, Murdoch University, Murdoch, Western Australia 6150, Australia; judy.dunlop@dbca.wa.gov.au; 5Department of Biodiversity, Conservation and Attractions, PO Box 51, Wanneroo, Western Australia 6946, Australia; colleen.sims@dbca.wa.gov.au (C.S.); keith.morris@dbca.wa.gov.au (K.M.); 6360 Environmental, 10 Bermondsey Street, West Leederville, Western Australia 6007, Australia; FelicityJones@360environmental.com.au

**Keywords:** burrowing bettong, genetic mixing, intraspecific hybridization, translocation

## Abstract

The use of multiple source populations provides a way to maximise genetic variation and reduce the impacts of inbreeding depression in newly established translocated populations. However, there is a risk that individuals from different source populations will not interbreed, leading to population structure and smaller effective population sizes than expected. Here, we investigate the genetic consequences of mixing two isolated, morphologically distinct island populations of boodies (*Bettongia lesueur*) in a translocation to mainland Australia over three generations. Using 18 microsatellite loci and the mitochondrial D-loop region, we monitored the released animals and their offspring between 2010 and 2013. Despite high levels of divergence between the two source populations (*F*_ST_ = 0.42 and ϕ_ST_ = 0.72), there was clear evidence of interbreeding between animals from different populations. However, interbreeding was non-random, with a significant bias towards crosses between the genetically smaller-sized Barrow Island males and the larger-sized Dorre Island females. This pattern of introgression was opposite to the expectation that male–male competition or female mate choice would favour larger males. This study shows how mixing diverged populations can bolster genetic variation in newly established mammal populations, but the ultimate outcome can be difficult to predict, highlighting the need for continued genetic monitoring to assess the long-term impacts of admixture.

## 1. Introduction

Species used in conservation translocations are often threatened or rare. Many have isolated populations that are subjected to loss of genetic variation, high levels of inbreeding, and elevated risks of extinction [1,2,3]. While it has been argued that any threatened populations with unique characteristics or distinct evolutionary history should be conserved separately [4], low genetic diversity and fitness reduction in small populations due to genetic drift and inbreeding can pose significant extinction threats to these populations [5]. Furthermore, an isolated population may accumulate deleterious mutations and have low evolutionary potential in changing environments [6]. For these reasons, translocation programmes involving single source populations may have a high risk of failure.

Mixing individuals from different source populations is one way to bolster genetic variation and avoid inbreeding in threatened species [7,8,9]. Hybridization between diverged populations counteracts the deleterious effects of inbreeding by masking deleterious recessives (dominance) or increasing heterozygosity at loci where heterozygotes have a selective advantage (over-dominance) [10]. A well-known example is the genetic restoration of the Florida panther (*Puma concolor coryi*). The introduction of Texas panthers (*Puma concolor stanleyana*) from a geographically nearby population increased genetic diversity, reduced inbreeding, improved survival and fitness, and tripled the number of panthers [11,12,13]. Furthermore, long-isolated populations often carry different subsets of alleles as a result of lack of gene flow, genetic drift, and local selection [14]. Interbreeding between individuals from these populations should increase evolutionary potential and enable their offspring to survive in a wider range of environments [15,16]. Consistent with this expectation, many translocations sourcing from multiple source populations or genetic rescue cases have shown an increase in genetic diversity and improved fitness over multiple generations [7,17,18,19,20,21].

While mixing differentiated populations can have favourable outcomes, it can also lead to fitness reductions in the progeny (i.e., outbreeding depression) due to post-zygotic isolation between source populations [22,23,24]. Crossing between phenotypically different parents can produce offspring with phenotypes that are unsuitable for the local environment. For example, a hybridisation of two garter snake populations (*Thamnophis ordinoides*) produced a mismatch in body pattern and behaviour in hybrid snakes that had a higher mortality from predation in comparison to purebred snakes [25]. Progeny may be unviable because of abnormal structure and/or number of chromosomes [26] or fitness of progeny may be lower due to heterozygote disadvantage, harmful epistatic interaction between alleles of the parents, or disrupting of co-adapted gene complexes [27]. Common signs of intrinsic incompatibility include reduction in fertility and viability of hybrid offspring, such as sterility [26], low survival rate [28], slow growth rate [29], and decreased reproductive success [30]. In addition, pre-zygotic isolation due to differences in morphology, behaviour, ecology, reproductive biology, and gametic compatibility, may prevent or reduce interbreeding between individuals from different source populations [31,32,33]. This can reduce the effective population size or result in an uneven genetic contribution from the source to the translocated population and induce genetic problems associated with a small population size [34]. 

Predicting whether outbreeding depression will occur is difficult. Generally, the risk of outbreeding depression becomes higher as the genetic distance between the parents becomes greater, but the amount of divergence required for it to occur varies from species to species [35,36]. The different possible outcomes of mixing diverged populations leave many conservation managers with a difficult decision when choosing populations for use in translocations. This decision can affect the outcome of the translocation and long-term persistence of the population. Allendorf et al. [24] and Edmands [22] suggested that augmenting gene flow between fragmented populations should only be carried out if the populations have lost substantial genetic variation and the effects of inbreeding depression are apparent. However, such information is often not available for populations of immediate conservation concern, and while awaiting data on the effects of inbreeding to be collected, populations are at risk of extirpation. Weeks et al. [37] argued that by overestimating the risk of outcrossing breeding depression, rational use of gene flow for genetic rescue is unnecessarily prevented. So far, a meta-analysis of intentional outcrossing of inbred populations of vertebrates, invertebrates, and plants with a low outbreeding depression risk (evaluated using Frankham et al. 2011 decision tree) has shown positive outcomes of genetic mixing [8], which has lasted for several generations [20]. However, there are only a few case studies available that employed outcrossing for conservation purposes [8,9]. Inconsistent outcomes of hybridisation between species, subspecies, and divergent populations and the increasing use of translocation mean that more studies are needed to allow better guidelines about when to use multiple populations in translocation [38].

The boodie or burrowing bettong (*Bettongia lesueur*) is a medium-sized, burrow-living marsupial endemic to Western Australia. *B. lesueur* is characterised by a short blunt head, with small rounded and erect ears. They are yellowy grey with a light grey underside, while the legs, feet, and tail are more yellow in colour [39] (Figure 1b). They are omnivorous, nocturnal, and the only macropod that shelters in burrows on a regular basis [39]. The average weight of a Bernier and Dorre Islands boodie is 1.26 ± 13.2 kg [40], in comparison to Barrow Island, which is 0.74 ± 9.2 kg [41] (Figure 1a). The populations on Bernier and Dorre Islands breed throughout the year, peaking over winter when the majority of rain falls [40]. On Barrow Island, breeding cycles are seasonally opposite to Bernier and Dorre islands and peak in summer, coinciding with cyclonic rain [40,41]. Females produce two young on average per year (three maximum) [40]. The young reach sexual maturity after 280 days [40]. Boodies form social groups of one male to one to many females [42]. Individuals in a social group can share multiple warrens, but males were observed to use more warrens than females [41,42]. There is no clear dimorphism between sexes [40]. Males gain access to females by investing in knowledge of the reproductive status and location of females within males’ day-range [43]. A male chases other males to defend females, but will often get supplanted, especially those with several females in their social group [42]. On the basis of the social behaviour and space used, they exhibit a polygynous mating system [42]. Boodies can survive to at least three years of age [40], but animals up to 11 years have been reported in a translocated population (J. Short et al., unpublished data).

The boodie is listed as Near Threatened in the 2019 International Union for Conservation of Nature (IUCN) Red List of Threatened Species [44] and as Threatened by the Environment Protection and Biodiversity Conservation (EPBC) Act (1999). Prior to European settlement, they were abundant throughout the middle and western half of Australia [45], but now only remain on Bernier, Dorre, and Barrow Islands. The disappearance of boodies from the mainland is likely to due to predation by foxes and feral cats [46]. Other factors contributing to their disappearance include competition with rabbits for food and shelter, habitat loss from land clearing for agriculture and livestock grazing, and changes in fire regimes [46]. *B. lesueur* on Bernier and Dorre Islands are considered to be a different subspecies to those on Barrow Island due to their significant body size differences [46] and a long period of isolation from mainland Australia (over 8000 years) [47]. Several reintroductions to nearby islands and to mainland sites have been carried out using individuals from one subspecies or the other [46]. However, only one reintroduction to a mainland site at the Matuwa Indigenous Protected Area (formerly Lorna Glen) has involved individuals from both subspecies. Matuwa Indigenous Protected Area is a large reserve of approximately 244,000 hectares. The reserve is actively managed by the Department of Biodiversity, Conservation and Attractions. For species sensitive predation by feral cats such as the boodie, 1100 hectares of introduced predator-proof fenced enclosure was constructed inside the reserve. Boodies from Barrow Island and Dryandra Field Breeding Facility (established from Dorre Island boodies) were released into this enclosure in 2010 [48].

According to the framework of Frankham et al. [49], the likelihood of outbreeding depression in this translocation is high, and it is unknown whether animals from different source populations will successfully interbreed because of their differences in size and long period of isolation. The main aim of this project is to confirm interbreeding and investigate the genetic consequences of mixing two geographically isolated source populations in the newly established translocated population of *B. lesueur* at Matuwa. Our specific aims were to first measure the genetic variation within and between source populations used in the Matuwa translocation with mitochondrial and nuclear markers. Second, we investigated the extent of genetic mixing between the two source populations, especially whether the body size difference has influenced the symmetry of introgression. Third, we measured levels of genetic variation to assess whether it was higher in the translocated population relative to the source populations, and whether levels of genetic variation have changed over time. Lastly, we investigated the genetic basis of phenotypic variation in body size between source populations and outcome of interbreeding on hybrid offspring body size. 

## 2. Materials and Methods 

### 2.1. Translocation Site and History

In 2010, 56 females and 53 males from Dryandra Field Breeding Facility (established from 20 individuals collected from Dorre Island in 1997, Dryandra hereafter) and 27 females and 40 males from Barrow Island were reintroduced to Matuwa (26°13′ S, 121°33′ E). All 67 Barrow Island founders were released in February, while Dryandra founders were released incrementally as follows: 20 in January, 80 in August (11 males and 11 females of these were recaptured two months later and moved outside the translocation site), and 9 in October.

### 2.2. Sampling and DNA Extraction

We collected samples during the establishment of the translocated population at Matuwa in 2010 and during follow-up population monitoring between 2010 and 2013. Six cohorts were sampled in total, two collected from individuals translocated to Matuwa (representing each of the source populations: Barrow Island, *N* = 67 and Dryandra, *N* = 109) and four collected from the population at the translocation site once a year from 2010 to 2013 (2010 *N* = 11, 2011 *N* = 27, 2012 *N* = 48, and 2013 *N* = 24). All animals caught during trapping sessions had a tissue biopsy sample taken from their ear (stored in 70% ethanol) and were microchip implanted, weighed, sexed, aged, and measured for head length and long pes length. DNA was extracted using a ‘salting-out’ method [50] with a modification of 10 mg/mL Proteinase K added to 300 µL TNES and incubated at 56 °C. All Matuwa samples were used for mitochondrial DNA (mtDNA) and microsatellite analysis. Additional mtDNA sequences were obtained from Barrow Island (*N* = 49), Dorre Island (*N* = 5), and Bernier Island (*N* = 5) populations collected between 1999 and 2001 (F. Donaldson, unpublished data) and were included in haplotype network analyses. 

### 2.3. mtDNA Control Region Sequences

The mitochondrial (mtDNA) D-loop region was amplified using primers L15999M and H16498M [51]. PCR was performed using the following parameters: after an initial denaturation at 94 °C for 2 min, 30 iterations of 94 °C for 35 s, 57 °C for 45 s, 72 °C for 1 min followed by a final extension of 10 min at 72 °C. Reactions were performed in 25 µL volumes that contained 20 ng of DNA, 2.5 μL of 10× buffer (Fisher Biotech, Thebarton, South Australia), 4 µL of 25 mM MgCl_2_, 0.5 µL of 10 mM dNTP2, 0.5 µL of 10 µM of each primer, 0.25 µL of 0.5 U Tth Taq polymerase, and 14.75 µL of dH_2_O. Products were sequenced in an ABI 3730 sequencer using a commercial service (Australian Genome Research Facility Ltd, Perth, Australia), edited using SEQUENCHER (Gene Codes Corporation, Ann Arbor, MI, USA), and aligned with CLUSTAL W using default parameters [52].

### 2.4. Microsatellites

We genotyped each individual at 18 microsatellite loci that were developed for other macropod species, including 12 loci described in Donaldson and Vercoe [53], 3 loci from *Petrogale assimilis* [54], 1 from *P. xanthopus* [55,56], and 2 from *Potorous longies* [57] (Appendix A). PCR reactions (volume 10 µL) were performed using a QIAGEN Multiplex PCR Kit with 10 ng of DNA and primer concentrations ranging from 0.04 to 0.4 µM (Appendix A). Amplifications were carried out using the following parameters: 15 min at 95 °C, followed by 35 cycles of 30 s at 94 °C, 90 s at different annealing temperatures as described in Appendix A, and 60 s at 72 °C, concluding with 30 min at 60 °C. PCR products were analysed in an ABI 3730 sequencer using a GeneScan-500 LIZ internal size standard and scored using GENEMARKER version 1.90 (SoftGenetics, Pennsylvania, USA).

### 2.5. Data Analysis

Mitochondrial DNA diversity was quantified by calculating the number of haplotypes, gene diversity and nucleotide diversity using DNASP version 5 [58]. Pairwise ϕ_ST_ values and tests for differentiation between samples taken from the source populations and translocation site were calculated and tested using an analysis of molecular variance (AMOVA) in ARLEQUIN version 3.0 [59]. Representatives of each haplotype were selected for haplotype network analysis. We constructed median-joining haplotype networks to visualise the relationships between haplotypes derived from different islands in NETWORK v5.0 [60]. For sites with deletions or insertions, we doubled the site’s weight from the default value (20), as suggested by the manual, and increased epsilon value to 10 as it generated a better network. Maximum-parsimony (MP) calculation was applied after the median joining calculation [61].

We assessed microsatellite genotyping quality by calculating the allele- and locus-specific genotypic error rates by re-genotyping randomly selected 10% of the total sample [62]. We tested the presence of null alleles in the source populations at each locus using MICROCHECKER [63]. Estimates of the allelic richness (an estimate of the number of alleles per locus corrected for sample size), gene diversity, inbreeding coefficient (*F*_IS_), pairwise genetic divergence (*F*_ST_), tests for differentiation among population samples, and genotypic disequilibrium were calculated using FSTAT version 2.9.3.2 [64]. The significant deviation of *F*_IS_ values from Hardy–Weinberg Equilibrium was determined by randomization tests. Genetic divergences between pairs of population samples were quantified using Weir and Cockerham’s [65] *F*_ST_ (*θ*). Genotypic disequilibrium between each pair of loci within each population sample was assessed by testing the significance of association between genotypes. For these tests, a sequential Bonferroni correction [66] was applied to control for type I statistical error. Differences in gene diversity and allelic richness among population samples were statistically tested using Wilcoxon’s signed-rank tests with samples paired by locus using the R version 3.5.3 statistical program [67].

Two methods were used to infer the extent of genetic mixing within the translocation site. Firstly, we used a Bayesian clustering method in STRUCTURE 2.3.4 [68]. Analyses were performed assuming the presence of two genetic clusters (*K* = 2) because this was the number of source populations. To confirm the number of genetic clusters, we compared the likelihood values for different *K* values (1–10) and used the ΔK method of Evanno et al. [69] to choose *K* (Appendix A). Ten independent runs were performed using 100,000 iterations, with a burn-in period of 10,000 iterations. These data were also used to determine whether the genetic composition of the translocated population differed to the expected genetic composition based on the numbers of founders from each source population. To do this, the average proportion of membership to each predefined cluster was compared to the expected proportions based on the number of individuals translocated using a chi-squared test. After taking into account individuals that were known or assumed to have died within one month of release, there were 59 founders from Barrow Island and 67 from Dryandra. Secondly, we used NewHybrids version 1.1 [70] to assign individuals born at the translocation site as one of the six generations. Genotype frequency classes of each generation were specified as follows: pure-bred Barrow Island (1.000/0.000/0.000/0.000), pure-bred Dryandra (0.000/0.000/0.000/1.000), F1 hybrid (0.000/0.500/0.500/0.000), backcross to pure-bred Barrow Island (0.500/0.250/0.250/0.000), and backcross to pure-bred Dryandra (0.000/0.250/0.250/0.500) [70]. The z option was set as prior information for individuals from the source populations. Uninformative priors (Jeffreys) were given to both allele frequency and admixture distributions. Five replicates were run for 1,000,000 Markov chain Monte Carlo (MCMC) sweeps following a burn-in period of 100,000. Individuals with a posterior probability value below 0.7 were excluded from further analyses. The results were cross-checked with STRUCTURE Q values. 

We examined the direction of mixing further by comparing the proportions of mtDNA haplotypes in F1 hybrids at the translocation site with the expected values based on the haplotype frequencies and number of females translocated from each of the source populations. The expected proportions were adjusted as described above to exclude females who were unlikely to contribute to the gene pool. We also carried out these comparisons for pure-bred Barrow Island and pure-bred Dryandra generations born at the translocation site to test whether haplotype frequencies in the offspring matched the haplotype frequencies of the female parents, on which the predicted values were based. Each generation was analysed separately and deviations between observed and expected numbers of haplotypes were tested using chi-squared analysis.

To investigate the relationship between body size measurements and ancestry, we compared the body weight, head length, and pes length of adults originating from the source populations or born at the translocation site. The measurements were taken from genotyped animals that were captured between 2010 and 2015. We compared body size measurements of individuals grouped by ancestry class: pure-bred Barrow Island, pure-bred Dryandra, and F1 hybrid to each source population determined using NewHybrids and STRUCTURE results. The significance of differences in body size measurements between pure-bred individuals born at the translocation site and between the source populations were determined using unpaired *t*-tests. 

## 3. Results

### 3.1. Mitochondrial DNA Variation

Fifty-three polymorphic sites were found in the 580-bp D-loop region, giving a total of fifteen haplotypes. These haplotypes were deposited in GenBank (Accession numbers: KP257602–KP257616). Four distinctive clades were apparent, corresponding to each of the remnant populations. Two clades were found within the Barrow Island population and one unique clade was detected in each of the Dorre and Bernier Island populations (Figure 2a). The AMOVA revealed 69.1% of the variation was between island populations.

Each of the source populations used to establish the Matuwa translocated population had both shared and unique haplotypes. Three haplotypes (A, B, and C) were found in both the Dryandra and Barrow Island population samples. Individuals originating from Barrow Island also carried unique haplotypes (D, E, G, and H), while individuals originating from Dryandra had only one unique haplotype (I) in addition to shared haplotypes (Figure 2b). Haplotypes F, J, K, L, M, N, and O were not found in any of the individuals used to establish the translocated population. Most of the individuals (90.1%) from Dryandra had the haplotype I, while most of the individuals originating from Barrow Island carried the haplotype A (57.5%, Figure 2b). 

Levels of mtDNA variation in the translocated population at Matuwa were comparable to the levels found in the Barrow Island source population but were higher than the Dryandra source population (Table 1). Significant differences in haplotype frequencies were detected between the source populations and between the source populations and samples taken from the translocation site (Table 2). There were also significant differences in haplotype frequencies between collection years within the translocated population. Temporal variation in haplotype frequencies were observed between 2010 and the other collection years only (Table 2).

### 3.2. Microsatellite Variation

Across the microsatellite data set, the amplification success rate was 0.973 per locus. The allele- and locus-specific genotypic error rates were 0.030 and 0.039, respectively. Six loci (Y151, Y170, Y76, Y112, T17-2, and Pa597) showed evidence of null alleles in the source population samples, but none were consistently found in both populations. The data were analysed with and without loci with null alleles. Both showed similar results, but the presence of null alleles lowered the posterior probability of some samples in NEWHYBRIDS. Therefore, only results without null alleles were presented. Overall, estimates of genetic diversity were typically higher in the translocated population at Matuwa than the source populations (Table 1). Pairwise tests revealed significantly higher allelic richness and gene diversity in each of the samples taken from the Matuwa translocation site when compared to the Dryandra source population (Wilcoxon rank sum tests, *P* < 0.05 in all cases). However, there were no significant differences between samples from the translocation site and the Barrow Island source population or between samples taken from the translocation site in different years.

All population samples from the translocation site had significantly positive multilocus *F*_IS_ values (randomization tests, *P* < 0.008). Multilocus *F*_IS_ values of the source populations were not significantly different from zero (Table 1). The number of pairs of loci in genotypic disequilibrium (GD) ranged from 0 to 24, with the highest levels occurring in the 2011 and 2012 population samples taken from the translocation site. The number of pairs of loci in GD in the source population samples ranged from 0 to 2 (Table 1).

### 3.3. Population Structure and Genetic Mixing

Pairwise population ϕ_ST_ and *F*_ST_ values based on mtDNA and microsatellite data, respectively, indicated there was substantial genetic differentiation between the source populations (Table 2). There were also significant divergences among the different collection years sampled from the translocation site, and between the translocation site and both source population samples. Initially, ϕ_ST_ and *F*_ST_ values indicated the translocated population was most similar to the Barrow Island source population. However, it became more similar to the Dryandra source population over time. This pattern was more pronounced in the mtDNA data.

The clustering analyses also revealed changes in the genetic composition of the translocated population over time (Figure 3). Of the 11 offspring born at the translocation site during 2010, 10 had an ancestry matching the founders from the Barrow Island population. There was also an individual with an ancestry matching the Dryandra source population. Offspring with mixed ancestry started to appear at the translocation site from March 2011. Genetic mixing between the source populations was also clearly evident in the 2012 and 2013 population samples based on observed cluster membership proportions. Overall, the proportion of membership to each predefined genetic cluster in the population samples from the translocation site was not significantly different from the predicted proportions based on the number of founders from each source population after taking early mortality into account (Table 3). 

Evidence of genetic mixing between source populations was also found with the NEWHYBRIDS analysis. Across all offspring born at the translocation site that had a posterior probability above the 0.7 threshold in NewHybrids (*N* = 109), 38.5% were designated as pure-bred Barrow Island, 28.4% pure-bred Dryandra, 18.3% F_1_ hybrid, and 14.7% post F_1_ generation. The classification in the NEWHYBRIDS analysis was consistent with STRUCTURE. All pure-bred individuals identified with NEWHYBRIDS had Q-values greater than 80% and all hybrid individuals had Q-values between 42% and 65%. Of the 20 F_1_ hybrids identified, 18 (90%) had a candidate mother from the Dryandra source population or was an adult female born at the translocation site with pure-bred Dryandra ancestry (Table 4). This pattern was supported by the mtDNA data, which revealed higher than expected numbers of Haplotype I, which was restricted to the Dryandra source population (Table 4). There were no significant differences between observed and expected haplotype frequencies in the pure-bred Barrow Island and pure-bred Dryandra offspring (Table 4). 

### 3.4. Variation in Body Size Among Parental and Admixed Generations

Both females and males from the Dryandra source population were significantly heavier and larger than those from the Barrow Island source population (Figure 4). These differences were also apparent in pure-bred offspring born at the translocation site (Figure 4), suggesting the body size differences between the source populations have an underlying genetic basis. Significant differences between Matuwa-born pure-bred Dryandra individuals and Dryandra born founders were found in body weight in both sexes (females: *t* = 4.8, *P* < 0.001; males: *t* = 4.7, *P* < 0.001) and head length in females (*t* = 4.2, *P* < 0.001). Matuwa-born F_1_ hybrids were significantly larger than Matuwa-born Barrow Island pure-bred individuals and were similar in size to the Matuwa-born pure-bred Dryandra individuals, except for Matuwa-born F_1_ females, which had shorter pes length than pure-bred Dryandra born females (Figure 4). 

## 4. Discussion

The risk of pre- or post-zygotic reproductive barriers remains a key concern when using multiple source populations in translocations, especially when remnant populations of threatened species are highly diverged [37,49]. Our study revealed that despite the significant morphological and genetic differences between remnant island populations of *B. lesueur*, animals from these populations were able to interbreed and produced viable offspring. However, interbreeding between source populations was non-random. Below, we discuss the processes that may have led to this pattern, as well as genetic and morphological consequences of introgression and their implications for conservation of other mammal species.

### 4.1. Phenotypic and Genetic Differentiation Between Island Populations

Substantial phenotypic and genetic differentiation were evident between Bernier, Dorre, and Barrow Island populations of *B. lesueur*. Each population formed a unique clade or clades in the haplotype network analysis of mtDNA data. They also formed discrete genetic clusters following Bayesian cluster analysis of microsatellite data and had large differences in allele and haplotype frequencies. These genetic divergences likely reflect the geographical isolation between the remnant island populations, which occurred at least 8000 to 10,000 years ago [42] and further support the need for recognizing these population as separate taxa [71].

The body size differences between populations also appear to have a genetic basis, as pure-bred offspring born at the translocation site maintained the body size differences observed between source populations, despite being raised in the same environment. Furthermore, intermediate trait values in F_1_ hybrids suggest that mainly additive genetic effects control variation in some body size traits (e.g., pes length), although other traits where trait values in the F_1_ hybrid were comparable to individuals with a pure-bred Dryandra ancestry (e.g., body weight and head length) suggest that dominance or maternal effects are also important. 

### 4.2. Non-Random Mating and Asymmetrical Introgression Between Source Population Lineages

Despite high levels of genetic differentiation between remnant island populations, *B. lesueur* translocated to a new site on the mainland were able to interbreed and produced viable offspring. Admixed individuals were evident in the 2011 collection year after all of the founders were released. The lack of genetic mixing prior to 2011 likely reflected the different release times of founding animals, with the majority of Barrow Island founders being released in February 2010 and most Dryandra founders released mid-August 2010. After three years, more than half of the offspring born at the translocation site were of hybrid or backcrossed origin. Nevertheless, significantly positive *F*_IS_ values were evident in each yearly collection taken from the translocated site, suggesting that while interbreeding was taking place, mating between individuals with different ancestries was non-random. By contrast, no deviations from random mating were observed in the source population samples.

Non-random mating was also evident from the asymmetrical introgression between source populations; crosses between smaller-sized Barrow Island males and larger-sized Dryandra females were significantly more common than expected. It is unlikely that the asymmetry was caused by different breeding cycles because our trapping records indicated no obvious differences in the frequency of pouch young carried by females originating from Barrow Island and Dryandra. Another possibility is that intrinsic, cyto-nuclear incompatibilities may have selected against hybrids resulting from crosses between Dryandra males and Barrow Island females during gestation [72,73]. The cyto-nuclear incompatibilities would need to be partial because we observed bi-directional rather than a unidirectional introgression. A third explanation is that it might be too costly for smaller Barrow Island females (pre- or post-parturition) to provide enough resources for large offspring that might result from breeding with large Dryandra males. Freegard et al. [74], who developed an age estimation growth curve for *B. lesueur*, suggested that maternal weight may play a role in developmental variations of *B. lesueur* pouch young of the same age due to larger females producing more milk, resulting in larger pouch young. On Dorre and Bernier Islands, *B. lesueur* females with a weight range of 1000–1600 g produced pouch young more frequently (from signs of lactation or trapped with pouch young) [40]. Smaller-size Barrow Island females may not produce sufficient amount of milk to accommodate larger-sized hybrid pouch young, leading to early mortality.

In translocations, there are factors other than genetics that can influence the observed extent and direction of introgression, such as the pattern of mortality, release time and the relative proportions of different founder sources, variance in reproductive success, and selection. Some mortality is expected in translocations [75,76]. In fact, early mortality has been found to affect as much as half of released animals (e.g., *Capra ibex* [77] and *Sphenodon guntheri* [78]) with the level of genetic diversity in the translocated population determined by admixture between the surviving individuals. Our study demonstrated that the observed genetic proportion reflected the proportion of released animals after taking early mortality into account. In addition, the lag in admixture in this study also reflected the influence of founder release time. Reproductive success among founding individuals can be influenced by male body size, genetic relatedness, variation at the major histocompatibility complex (MHC), and the order of mating in the case of a polygamous mating system [79,80,81,82]. Mate choice can strongly influence introgression and result in unexpected outcomes, especially if the source populations exhibit morphological differences. For example, upon secondary contact between two species of Neotropical shorebird, *Jacana spinosa* and *J. jacana*, female–female competition is a major determinant of reproductive success [83]. Larger and more aggressive *J. spinosa* females disproportionally produce hybrid offspring, which subsequently shifts average female body mass to be more similar to larger-sized species [83]. Dominancy of one lineage can also lead to genetic swamping and/or increased inbreeding as a result of a reduction in the effective population size [34]. Lastly, selection will further drive changes in traits, which may dramatically change the genetic composition of the population over time [21].

### 4.3. Increased Genetic Diversity in a Newly Established Population

As found in previous studies on translocated populations involving multiple source populations, we found evidence of higher levels of genetic variation in the translocated population than in one or more of the source populations [7,18,84,85]. This result is not unexpected, as long-isolated populations often carry different subsets of alleles as a result of lack of gene flow, genetic drift, and local selection [14]. In addition, the large number of founders and high reproductive rate may have limited loss of genetic variation when the population was established [86]. Interestingly, the genetic diversity in the translocated population was not much greater than the level found in the most variable source population, Barrow Island. This result parallels Huff et al. [84], who found all reintroduced populations of slimy sculpins (*Cottus cognatus*) exhibited increased levels of genetic diversity, but the increases were only slightly higher than the single most genetically diverse source population. While we found no evidence of differences in genetic diversity among collection years, previous studies have shown that translocated populations often lose genetic diversity as a result of the founder effect [86], mating system [19,87], or survivorship differences of either founders or offspring [72,77,88]. If the asymmetrical introgression continues past the first few generations of mixing, it is possible that the translocated population may undergo significant genetic changes and lose genetic variation (see Rick et al. [89] in this issue for an analysis on the long-term impacts of introgression and the survivorship and reproductive fitness of different generational classes in this population).

### 4.4. Future Direction

There is little information on managing unexpected negative outcomes of introgression in wild populations, for example, outbreeding depression, genetic swamping, maladaptation, and unexpected loss of genetic diversity. In the case of outbreeding depression, two management strategies have been proposed to mitigate its deleterious effects: the selective removal of admixed individuals and the introduction of one parental stock to encourage backcrossing [37]. While these are potential solutions, they have never been tested and it is still unknown if a wild population will respond to such management intervention. An alternate option is to allow selection to remove incompatible gene interactions naturally. However, for selection to act effectively, this would require large population size (>1000) [37]. In many cases, especially those in enclosures and on islands, this cannot be achieved, and some species may not be able to rapidly recover to large population size (>1000) because of either intrinsic or extrinsic constraints. Therefore, continuous monitoring of such populations is necessary. Additional studies, such as detection of genetic markers under selection (with potential adaptive roles) through next generation sequencing in addition to neutral molecular markers, may be employed as another method to evaluate the success of translocations. More studies are needed to understand the effects of introgression at the genomic level (e.g., adaptive introgression) and testing effectiveness of different management strategies so that guidelines can be developed to manage unexpected outcomes of introgression in translocations, particularly if augmenting gene flow is to be used more frequently as a conservation management tool. 

## Figures and Tables

**Figure 1 genes-10-00729-f001:**
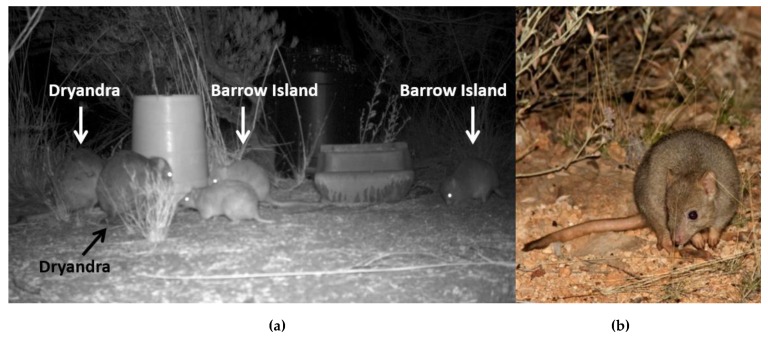
Body size differences between (**a**) *Bettongia lesueur* from Dryandra (originally Dorre Island) and Barrow Island from a camera trap taken at the Matuwa translocation site, and (**b**) *B. lesueur* in colour.

**Figure 2 genes-10-00729-f002:**
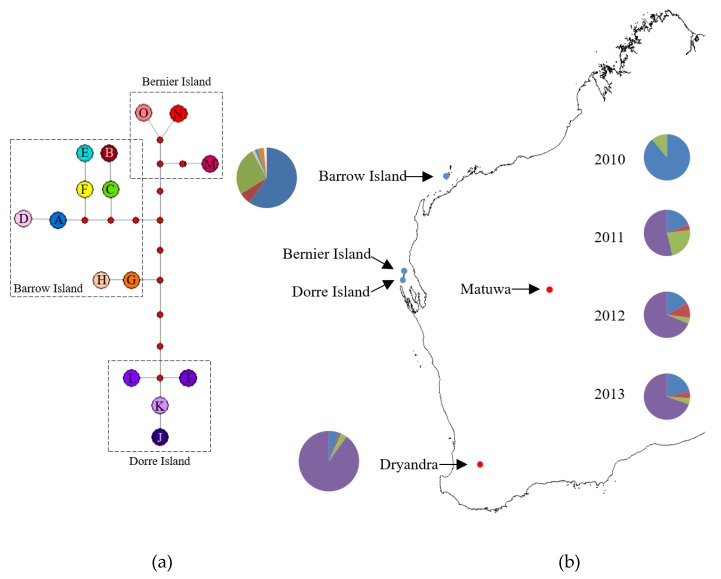
Haplotype network of *Bettongia lesueur* island populations and present distribution of *B. lesueur*. (**a**) Haplotype Network of 15 unique haplotypes detected in D-loop region of mitochondrial DNA. Boxes represent island populations. Each circle represents one haplotype with haplotype ID in the circle. Red dots and lines represent unsampled ancestral haplotypes and mutation steps, respectively. (**b**) Blue dots represent the locations of remaining natural populations (Barrow, Bernier, and Dorre Islands). Red dots are two translocated populations (Dryandra and Matuwa). Pie charts on the left represent haplotype frequencies in Barrow Island and Dryandra individuals used to establish the Matuwa translocated population. Pie charts on the right represent haplotype frequencies in samples taken from the Matuwa population between 2010 and 2013. Colours in pie charts represent haplotypes identified in (**a**).

**Figure 3 genes-10-00729-f003:**
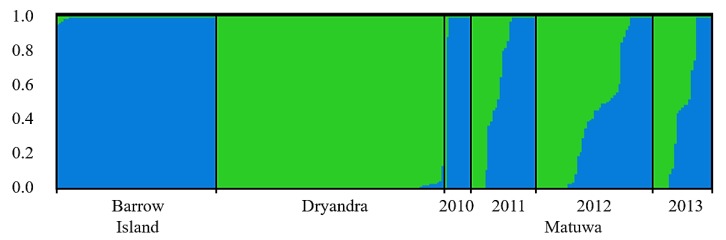
Summary of the Bayesian clustering results for the Matuwa translocated population assuming two admixed populations (*K* = 2). Each individual is represented by a bar showing its estimated membership to a particular cluster (represent by different colours). Black lines separate samples from different source populations (Barrow Island and Dryandra) and collection years at the Matuwa translocation site.

**Figure 4 genes-10-00729-f004:**
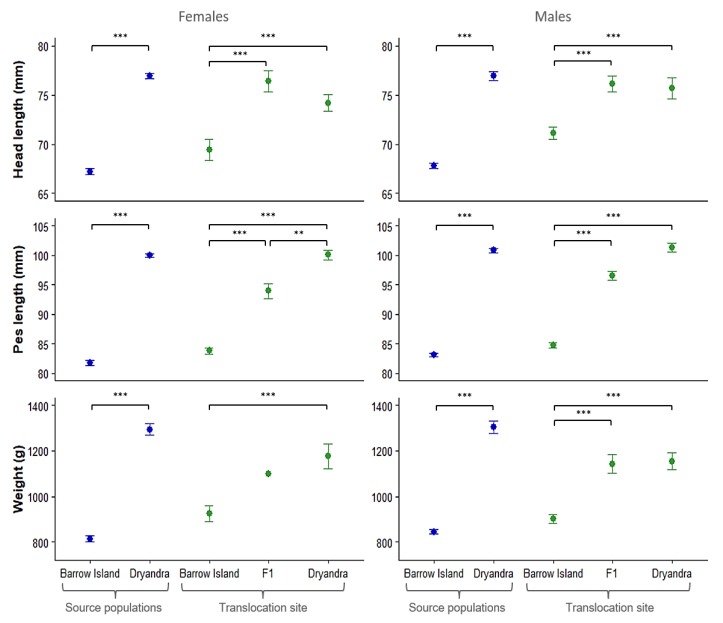
Mean body size in the source populations (blue) and different generations born at the Matuwa translocation site (green). Error bars are standard errors. Significant levels: * *P* < 0.05; ** *P* < 0.01; *** *P* < 0.001.

**Table 1 genes-10-00729-t001:** Sample sizes, estimates of genetic variation, inbreeding coefficient (*F*_IS_), and genotypic disequilibrium (GD) in the source (Barrow Island and Dryandra) and translocated populations (Matuwa) based on 12 microsatellite loci. *N* is the number of samples used in analysis. *H* is gene diversity. Standard errors are given in brackets after mean values. *F*_IS_ estimates significantly greater than 0 after correction for multiple comparisons are denoted with an asterisk.

Population	Microsatellites					Mitochondrial DNA			
	*N*	Allelic Richness	*H*	*F* _IS_	Pairs of Loci in GD	*N*	Number of Haplotypes	*H*	Nucleotide Diversity
Barrow Island	66	3.8 (0.5)	0.56 (0.07)	0.02	0	62	7	0.58 (0.01)	0.008 (0.0003)
Dryandra	95	2.8 (0.2)	0.47 (0.07)	−0.03	2	91	4	0.19 (0.01)	0.004 (0.0001)
Matuwa 2010	11	4.4 (0.5)	0.61 (0.05)	0.18 *	0	9	2	0.22 (0.06)	0.001 (0.0004)
Matuwa 2011	27	4.9 (0.5)	0.69 (0.04)	0.19 *	16	26	4	0.64 (0.01)	0.012 (0.0002)
Matuwa 2012	48	5.0 (0.5)	0.69 (0.05)	0.11 *	24	48	4	0.50 (0.01)	0.010 (0.0002)
Matuwa 2013	24	4.9 (0.5)	0.71 (0.04)	0.19 *	3	24	4	0.47 (0.02)	0.009 (0.0004)
Matuwa overall	113	5.1 (0.5)	0.70 (0.04)	0.17 *	64	110	4	0.59 (0.00)	0.011 (0.0001)

**Table 2 genes-10-00729-t002:** Pairwise *F*_ST_ and ϕ_ST_ values between the source populations and samples collected from the translocated population (Matuwa) between 2010 and 2013, based on microsatellite (below diagonal) and mitochondrial DNA (mtDNA) data (above diagonal). Significant *F*_ST_ values after correction for multiple comparisons and ϕ_ST_ values are denoted with an asterisk.

	Barrow Island	Dryandra	2010	2011	2012	2013
Barrow Island	–	0.72 ^*^	−0.02	0.35 ^*^	0.49 ^*^	0.50 ^*^
Dryandra	0.42 ^*^	–	0.75 ^*^	0.30 ^*^	0.11 ^*^	0.10
2010	0.01	0.38 ^*^	–	0.26 ^*^	0.43 ^*^	0.44 ^*^
2011	0.09 ^*^	0.20 ^*^	0.04	–	0.02	0.02
2012	0.18 ^*^	0.10 ^*^	0.11 ^*^	0.02	–	–0.03
2013	0.17 ^*^	0.13 ^*^	0.10 ^*^	0.02	0.00	–

**Table 3 genes-10-00729-t003:** Observed and the initial expected proportions of membership to each predefined cluster in samples taken from the Matuwa population. Observed proportions are based on mean Q-values from the STRUCTURE analysis of microsatellite data. Expected proportions are based on the number of individuals translocated from each source populations after individuals with known or assumed mortality were removed. For convenience, clusters have been labelled according to the source population they define (BWI = Barrow Island and DRY = Dryandra).

Sample	*N*	Observed		Expected		χ^2^	*P*
		BWI	DRY	BWI	DRY		
2010	11	0.895	0.105	0.468	0.532	–	–
2011	27	0.609	0.391	0.468	0.532	2.1	0.143
2012	48	0.431	0.569	0.468	0.532	0.3	0.605
2013	24	0.450	0.550	0.468	0.532	0.0	0.858
Overall	113	0.537	0.463	0.468	0.532	2.1	0.143

**Table 4 genes-10-00729-t004:** Observed and expected numbers of haplotypes in individuals born at the Matuwa translocation site. Individuals were classified as pure Barrow Island, pure Dryandra, or F_1_ hybrid, based on the results from the NEWHYBRIDS analysis. Expected numbers for each haplotype were based on the haplotype frequencies and the number of females translocated from each source populations after females with known or assumed mortality were removed. Haplotypes with low expected numbers were pooled prior to carrying out the chi-squared test.

Sample	Haplotype					χ^2^	*P*
	A	B	C	I	Others		
Pure Barrow Island	23 (26.7)	6 (1.9)	11 (9.5)	–	0 (1.9)	1.99	0.158
Pure Dryandra	1 (1.2)	0 (0.0)	0 (1.2)	30 (28.5)	–	0.96	0.327
F_1_ hybrid	1 (5.7)	1 (0.4)	0 (2.4)	18 (11.2)	0 (0.4)	9.48	0.002

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
