# Peer review of "Mixing Genetically and Morphologically Distinct Populations in Translocations: Asymmetrical Introgression in A Newly Established Population of the Boodie (Bettongia lesueur)"

_genes, 2019, doi:10.3390/genes10090729_

Round 1

Reviewer 1 Report

The manuscript entitled Mixing genetically and morphologically distinct populations in translocations: Asymmetrical introgression in a newly established population of the boodie (Bettongia lesueur)” shows an interesting research on a small marsupial species translocated among different Australian areas. This study can represent a good picture on the genetic and biological consequences of mixing two different populations in translocation programs. Although it is overall good in the development of the work, the manuscript has several confusing points that deserve to be clarified. In particular, I highlight the main points here:

I found very difficult to follow the names of the islands/populations/locations, because they change throughout the text and figures/tables. For example, in the abstract and in the introduction the Shark Bay Islands population are presented as one of the two source populations; afterwards figures and tables plus the text refers to the Dryandra source population (that was established from the Shark bay one). Moreover, at line 98 it is written that the Bernier and Dorre islands would henceforth be called Shark bay, but then the individual names Dorre and Bernier return. I suggest to uniform the names and simplify them. If you want to show a phylogeographic representation of the mtDNA haplotypes distribution, you can choose a tree or a network (it could also be an idea). If you prefer the tree, it might be useful to calculate with the Bayesian approach the time of divergence of the main clades as well, to verify if the genetic divergence between the two groups is the one hypothesized at lines 388-390. This tree can be a figure in the text, not a supplementary. I found unusual to use insular individuals to restock mainland populations (could you add a reference for this situation?), therefore it is good to try to understand when they diverged and the reasons of the extinction in continental Australia, compared to the healthy and numerous populations on the islands. Factors like predations, human interference, etc. could recur in the future… Where they taken into consideration when restocking was planned? Line 25 of the abstract: how many generations of boodies? The name of the species Bettongia lesueur can be shortened in lesueur throughout the text and in the captions after the first time it is named. In Figure 1, please put the (a) and (b) inside the right picture. Lines 101-102. The authors state that other translocations have been made before from the two islands to mainland Australia. Why hasn't a genetic analysis of the animals involved been planned before? And these founder animals are still there or not? Lines 115-117. I think that the genetic basis of phenotypic variation in body size (e.g. the genes involved or the functional pathway) was not found in this research, so here you can simply describe population variability in neutral genetic markers. Lines 123-124: please add standard deviations to the average weights. Lines 125-126: The opposite seasonal periods for the reproduction of the two populations then changed when the animals of the two populations were together on the continental site? Paragraph 2.1 does not deal with materials and methods, but is a description of the species (to be moved into the introduction). Where is paragraph 2.2? Lines 155-156: how did you avoid pseudoreplication (the re-sampling of the same animals)? Do you recognize the marks on the ear? You have different sample sizes for the mtDNA sequences obtained from island populations and in different years of Matuwa sampling: it’s important to keep this point into consideration. Another difference as regards sample size is in the number of individuals reintroduced from the two islands. When discussing the hypotheses for the female choice of a small male reproducer, you can also consider these numbers. At lines 147-148 the authors state that 20 of the Dryandra founders were removed two months later, how many males/females? In Materials and Methods, at line 210-213, simulations on the most probable number of genetic clusters (K) is announced. The Table S2 presents the results, but in the text of paragraph 3.3 and in Figure 3, the final choice and assumption of K=2 are not explained. Table 1. Why the column of N (number of samples used in the analysis) has standard deviations? To make Table 1 more readable, perhaps it is better to put the two values of each items in two lines or other settings like this. The second hypothesis for the asymmetry in mating (the cyto-nuclear incompatibility at lines 414-417) does not seem so probable, as the authors also mention. I was wondering if another possibility could be that a small female with an “oversized” son may have mechanical problems during gestation or at birth (post-zygotic barriers). I don’t know if there are clues of this kind for mother/son death or abortion… Does the article at line 460 (Rick et al.) has been accepted for publication? In the “future direction” section, I think it is premature to talk about how to mitigate negative outcomes of outbreeding depression, etc. First of all, it is important to know the genetic composition of a population BEFORE restocking, so I would suggest to check it when planning such actions. Then, for an animal species with such short generation times, it is essential to CONTINUE genetic monitoring over the years (in this work the situation is updated to 2013, six years ago). Finally, to effectively evaluate the success of reintroductions and translocations, the adoption of genetic markers under selection (with a potential adaptive role) is desirable, in addition to the study of neutral molecular markers. References: in many articles the animal species have the genus name in lowercase (e.g. ovis canadensis, thamnophis ordinoides, ). Please correct. Moreover, some Journals are abbreviated, some others not. Please uniform. Table S1: the annealing Temperature of some loci in the same multiplex are different (Y170 at 62° and Pa593 at 60° in the third multiplex, which goes to 57°C), is it right? Check also the references of Tab S1. e.g. reference N°51 refers to the Clustal software. Although the primers used have already been published, this is the first description of their application on this animal species. It may be useful for other researchers to also have information on allele frequencies as supplementary files. However, if the authors are not happy with this, I don’t want to insist on this point.

In summary, this is an interesting experimental research on the genetic outcomes of translocation programs. In my opinion, it need some changes to clarify the manuscript.

Reviewer 2 Report

Line 125 in winter or over winter not 'in over'

I think there is a need for some more information about the release site and the population history post release. 

1) How big was the enclosure into which the animals were released?

2) Do we have population history at the introduction site a) births b) deaths during each year from the introduction to the end of the study.

3) Do we have any information about the population history after the introduction period studied. i.e. did the population survive and increase during the period 2013-2019. I believe this information is available from annual reports of the Matuwa reserve. i.e. http://gemg.org.au/ckfinder/userfiles/files/Matuwa%20Annual%20Report%202016.pdf
